# TalkUp: Paving the Way for Understanding Empowering Language

**Lucille Njoo**♣* **Chan Young Park**◇* **Octavia Stappart**♣
**Marvin Thielk** **Yi Chu** **Yulia Tsvetkov**♣
♣ University of Washington ◇ Carnegie Mellon University
{lnjoo,ostapp,yuliats}@cs.washington.edu chanyoun@cs.cmu.edu
{marvin.thielk,yirosie}@gmail.com

## Abstract

Empowering language is important in many real-world contexts, from education to workplace dynamics to healthcare. Though language technologies are growing more prevalent in these contexts, empowerment has seldom been studied in NLP, and moreover, it is inherently challenging to operationalize because of its implicit nature. This work builds from linguistic and social psychology literature to explore what characterizes empowering language. We then crowdsource a novel dataset of Reddit posts labeled for empowerment, reasons why these posts are empowering to readers, and the social relationships between posters and readers. Our preliminary analyses show that this dataset, which we call TalkUp, can be used to train language models that capture empowering and disempowering language. More broadly, TalkUp provides an avenue to explore implication, presuppositions, and how social context influences the meaning of language.[1]

## 1 Introduction

Empowerment – the act of supporting someone's ability to make their own decisions, create change, and improve their lives – is a goal in many social interactions. For instance, teachers aim to empower their students, social workers aim to empower their clients, and politicians aim to empower their supporters. A growing body of psychology and linguistics research shows how empowerment – and disempowerment – can impact people by increasing their sense of self-efficacy and self-esteem (Chamberlin, 1997; Osborne, 1994).

Understanding how empowerment is conveyed in language becomes more important as language technologies are increasingly being used in interactive contexts like education (Molnár and Szüts, 2018), workplace communication (Prabhakaran

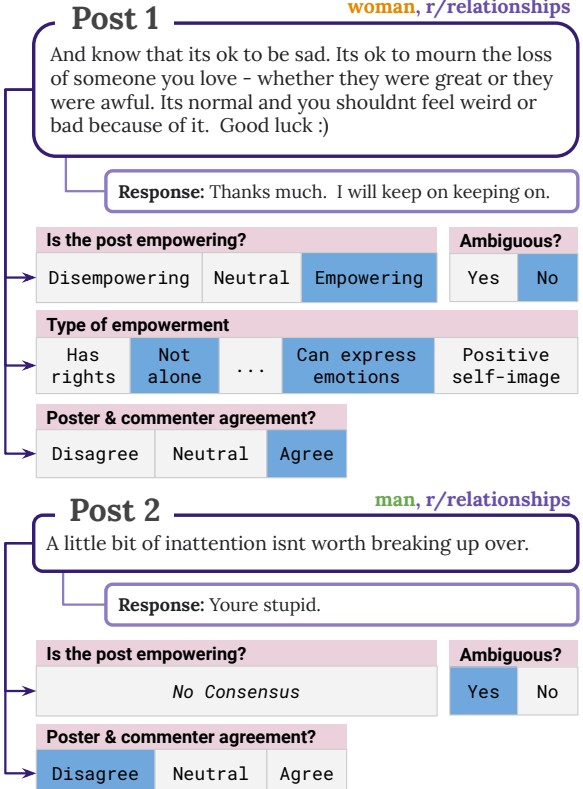

Figure 1: Two examples of annotated conversations in TalkUp. Post 1 is straightforwardly empowering, but Post 2 is inherently ambiguous and could either be interpreted as helpful advice or as a dismissive, belittling comment. Social context can also affect Post 2's implications: the post might elicit different reactions if it were written by a woman to a man or vice versa.

and Rambow, 2014a; Prabhakaran et al., 2012), and healthcare (Locke et al., 2021; Sharma et al., 2021a). Whether we are building dialogue agents for mental health support, supplementing children's education, or analyzing managers' feedback to their employees, language that empowers or disempowers the reader can have drastically different effects.

With a few exceptions (Ziems et al., 2022; Sharma et al., 2023), prior NLP research has focused on flagging *harmful* text, but there has been

---

*Equal contribution
[1]The data, codes, task instructions, and annotation UI can be found at https://github.com/chan0park/TalkUp.

much less investigation of what makes text *helpful*. Other works have studied related concepts like condescension (Wang and Potts, 2019) and implicit toxicity (Breitfeller et al., 2019a; Sap et al., 2020; Upadhyay et al., 2022), and we build off of these to construct a dataset that *complements* those tasks.

Consider the two examples of potentially empowering interactions in Figure 1. Empowerment exhibits the importance of social context in understanding the pragmatics of language: whether an exchange is interpreted as empowering or disempowering may depend on the participants' social roles and the power dynamics implied by their identities, including race, age, socioeconomic class, and many other social dimensions. Furthermore, empowerment cannot be easily detected with sentiment or emotion analyzers, since interactions with negative implicatures can be empowering (e.g., *you can quit!!!*), and messages that are positive on the surface can be disempowering (e.g., *you are so articulate for a girl!*) (Field and Tsvetkov, 2020). Modern language technologies do not model social context or deeper pragmatic phenomena, and thus are unable to capture or control for empowerment.

This work makes concrete steps towards understanding these linguistic phenomena by investigating the following research questions: **[RQ1]** What makes language empowering, and how is it manifested in language? **[RQ2]** Can empowerment be detected with computational approaches?

Our contributions are threefold: (1) We introduce the new task of empowerment detection, grounding it in linguistic and psychology literature. (2) We create TalkUp, a novel dataset of Reddit posts labeled for empowerment, the fine-grained type of empowerment felt by the reader, and the social relationships between posters and readers. (3) We analyze the data and demonstrate how it can be used to train models that can capture empowering and disempowering language and to answer questions about human behavior.

Ultimately, TalkUp aims to assist future researchers in developing models that can detect, generate, and control for empowerment, and to facilitate broader exploration of pragmatics. We have by no means covered every possible social dimension, but by focusing on a few social factors in the simplified setting of two-turn dialogues, we hope that TalkUp's framework can make strides toward understanding language in more complex social interactions, such as conversations involving intersectionality as well as longer multi-turn dialogues.

## 2 Background

We discuss empowerment following its definitions in clinical psychology (Chamberlin, 1997). We find this most appropriate for studying language because clinical psychology practice is usually centered around dialogue between clinician and patient, and because it involves concrete implications about individuals rather than vague cultural phenomena. Thus, summarizing the different characteristics of empowerment described in psychology literature, we define empowering text as *text that supports the reader's rights, choices, self-fulfillment, or self-esteem*.

Incorporating empowerment in dialogue agents, mental health support chatbots, educational assistants, and other social-oriented NLP applications is clearly a desirable goal. However, empowerment is inherently challenging to operationalize for several reasons. First, it is a flexible term that describes a wide range of behaviors across many domains – empowerment in economics, for example, looks very different from empowerment in a therapy session (McWhirter, 1991). We follow recent literature outside of NLP in trying to distill these varied interactions into a concrete definition. Second, empowerment is implicit: it is often read in between the lines rather than declared explicitly. Text might be empowering by reminding someone of their range of options to choose from, encouraging them to take action, asking for and valuing their opinion, or even validating their feelings (Chamberlin, 1997). Third, empowerment is heavily dependent on social context: whether or not a person is empowered depends on who is saying what to whom. We incorporate these consideration in our data collection process described next.

## 3 The TalkUp Dataset

We now discuss the TalkUp dataset's construction.

**Annotation Scheme**   Our annotation task[2] was shaped through multiple pilot studies, where we learned that context is useful for judging a post, annotators' free-response descriptions of social roles lack consistency, and posts are often inherently ambiguous. We elaborate on these findings in Appendix D. Based on these insights, the final task,

---

[2]Screenshots of user interface and full guidelines including definitions and examples of each label are in Appendix F.

which is illustrated in Figure 1, consists of three main parts:

(1) *Rating the post on an empowerment scale.* This scale has "empowering" on one end, "disempowering" one the other, and "neutral" in the middle. We define text to be empowering if it supports the reader's rights, choices, self-fulfillment, or self-esteem, and disempowering if it actively denies or discourages these things. Notably, posts that discuss an external topic without making any implications about the conversants, such as a comment about a celebrity's lifestyle, are defined as neutral.

(2) *Selecting why a post is empowering or disempowering.* We adopt the 15 points from Chamberlin (1997), with slight modifications to adapt them to written text, as *reasons why a post can be empowering* to a reader. Refer to Appendix E for the complete list of 15 reasons and corresponding definitions provided to annotators. If a post is empowering, it should imply one or more of these reasons (e.g. that the reader is capable of creating change), and if it is disempowering, it should imply the opposite (e.g. that the reader is not capable of creating change).

(3) *Selecting whether the poster and commenter have agreeing or disagreeing stances.* We define "agreeing" and "disagreeing" loosely in order to accommodate a wide range of social relationships: "agree" means that the poster and reader support the same point of view on a topic, whether it be politics, sports teams, or music preferences. "Disagree" means that they take opposing sides.

**Data Source** TalkUp consists of English Reddit posts from RtGender (Voigt et al., 2018), a collection of 25M comments on posts from five different domains, each labeled with the genders of the commenter and the original poster. We take advantage of the fact that these conversations are already annotated for gender, which provides contextual information about who is speaking to whom and allows us to explore at least one dimension of social context. [3]

Though RtGender contains posts from several platforms, given our focus on *conversational* language, we specifically selected RtGender posts

from Reddit because they were the most generalizable and contained natural-sounding conversations. We manually chose five subreddits, aiming to include (1) a diverse range of topics and user demographics, and (2) discussions that are personal rather than about external events unrelated to the conversants. The subreddits are listed in Table 1.

We filtered data from these subreddits to exclude posts or responses that exceeded 4 sentences in length or were shorter than 5 words. Additionally, we excluded posts with "Redditisms", and posts that were edited after they were initially posted (marked "EDIT:" by the original poster) and posts that began with quoted text (marked ">") were removed.

From pilot studies, we found that models can help to surface potentially empowering posts and help increase the yield of posts that were actually labeled as empowering by annotators. We trained a RoBERTa-based regression model with the data we collected from the pilot studies to predict the level of empowerment (0 for disempowering, 0.5 for neutral, 1 for empowering) in Reddit posts. We used this model to rank and select the top-k posts for annotation, and continually updated the model as we collected more data.[4] To ensure we annotate a diverse range of posts, our final annotation task was done with half model-surfaced posts and half randomly-sampled posts.

**Annotation on Amazon Mechanical Turk** With 1k model-surfaced posts and 1k randomly-sampled posts spread evenly among the five subreddits, we collected annotations via Amazon Mechanical Turk (AMT). Appendix F shows a screenshot of the user interface displayed to annotators. Each example was annotated by 3 different workers.

To ensure high quality annotations, we required annotators to have AMT's Masters Qualification,[5] a task approval rate of at least 95%, and a minimum of 100 prior tasks completed. Additionally, since our task requires English fluency, we limited annotators to those located in the US or Canada. Workers were compensated at $15/hour, and we calculated the reward per task based on the average time spent on each annotation in our pilot studies.

Following best practices to increase annotator diversity (Casey et al., 2017), we staggered batches

---

[3]We only consider men and women here due to the availability of data. We were not able to find any corpora that included nonbinary genders, but this is an important area for future work. Though we focus on gender, there are many other social variables that may impact empowerment, such as race and socioeconomic status.

[4]More information on the sample selection model is provided in Appendix B.1.

[5]AMT grants the "Master Worker" qualification to highly reliable workers.

|                     | Size | #E  | #D  | #A  | %W |
|---------------------|------|-----|-----|-----|-----|
| TalkUp              | 2000 | 962 | 129 | 267 | 43 |
| AskReddit           | 400  | 186 | 26  | 43  | 47 |
| relationships       | 400  | 199 | 35  | 83  | 72 |
| Fitness             | 400  | 193 | 28  | 64  | 14 |
| teenagers           | 400  | 173 | 29  | 48  | 34 |
| CasualConversation  | 400  | 211 | 11  | 29  | 50 |

Table 1: Data Statistics of TalkUp and breakdown of five subreddits in the data. #E: number of empowering examples, #D: number of disempowering examples, #A: number of ambiguous exmaples, %W: percentage of women posters in the data.

of data to be released at different times of day over multiple days. After each batch was completed, we manually quality-checked the responses and computed each annotator's standard deviation. We discarded data from unreliable annotators, including those who straightlined through many annotations with the same answer, those who clearly had not read instructions, and those whose alignment scores were more than 2 standard deviations from the mean. Annotator alignment scores were calculated by dividing the number of disagreements by the number of agreements between their label and the majority vote. We subsequently released new batches to re-label data previously annotated by the identified unreliable annotators.

**Data Statistics** We combined the *maybe empowering* with the *empowering* label, and did the same for the *disempowering* labels. We then used majority voting to aggregate the three annotations into the final labels for empowerment, ambiguity, and stance for each post. When all three annotators disagreed on the empowerment label (i.e., one vote each for empowering, neutral, and disempowering), we marked it as *No Consensus* and considered it an ambiguous case. For reason labels, where annotators can mark more than one categories per example, we only kept the reason labels that were marked by at least two annotators.

Table 1 shows the overall size of our dataset and the distribution of labels, the number of ambiguous cases, and percentage of posts made by women across the entire dataset and also by different subreddits. We annotated 400 posts from 5 different subreddits resulting in a total of 2000 samples. Of these, 962 were labeled as empowering, 129 as disempowering, and 267 as ambiguous, with 642 being labeled as neutral. We note that 265 out of the 962 empowering cases had no final

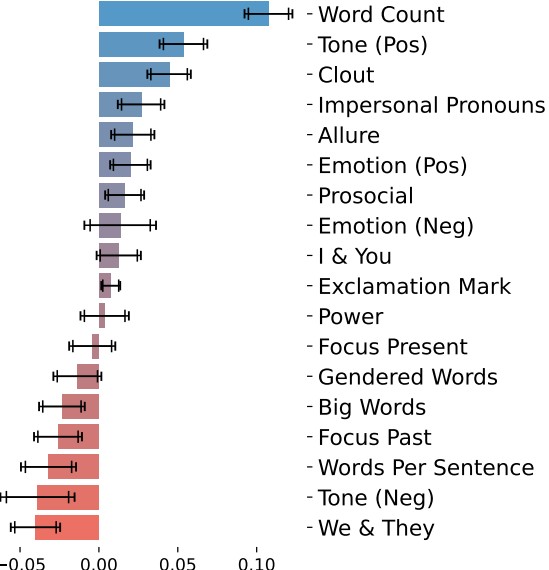

Figure 2: Weights of LIWC features with 90% and 95% confidence intervals assigned by linear regression model trained with TalkUp. All features except for Negative Emotion, Power, Focus Present have statistically significant weights ($p < 0.1$).

reason marked, indicating that there was no reason category annotators agreed on.

The inter-annotator agreement, Krippendorff's alpha, was 0.457, and the percentage agreement was 65.2%. These agreement scores are reasonable given the complexity and nuance of this task – we would neither expect nor want to have perfect annotator agreement because it is an inherently ambiguous problem even for humans, and there is often no objective "ground truth" on whether a text is empowering or not. Our agreement scores are comparable to those of other computational social science papers on tasks of similar nature, especially when concerning pragmatics. For example, our percentage agreement is higher than that of ElSherief et al. (2021)'s dataset on latent hatred, and our Fleiss's kappa is similar to that of the Microaggression dataset (Breitfeller et al., 2019b).

## 4 Data Analysis

We present preliminary analyses of TalkUp. Empowerment is a nuanced phenomenon in pragmatics and deeper exploration of social and linguistic variables remains open for future work. The analyses we present here provide some initial, surface-level insights into what makes language empowering.

## 4.1 Characteristics of Empowering Language

We use the LIWC-22 software to compute LIWC features for all annotated posts (Boyd et al., 2022). These features measure the percentage of word overlap between the text and predefined lexicons that capture different social and psychological characteristics of language, such as prosocial words or words associated with positive tone. For a more concise and generalized analysis, some related features are combined into compound features: the *I* and *You* features are grouped into one feature *I+You*, *We* and *They* into *We+They*[6], and *male* and *female* into *gendered words*. We standardize LIWC feature scores using the mean and variance calculated from TalkUp's randomly sampled posts. Model-surfaced posts are excluded as they may not reflect the distribution of Reddit posts in the wild.

To understand how each of these features contributes to empowerment in language, we train a linear regression model to predict the likelihood of a post being empowering. Figure 2 shows the regression coefficients assigned to each feature. Looking at the positive coefficients reveals that empowerment is associated with lexical features like *clout*, *allure*, *prosocial words*, and *exclamation marks*. Meanwhile, disempowerment is associated with features that have negative coefficients, such as big words and words-per-sentence, which may indicate sentence complexity. We expand on a few of the most notable findings below.

**Tone vs. Emotion.** We find that the *tone* of language is more influential to empowerment than the *emotion* conveyed. Positive tone has a significantly higher coefficient than positive emotion; likewise, negative tone is highly associated with disempowerment, while negative emotion is not statistically significant. This suggests that the concept of empowerment is distinct from sentiment and cannot be captured by sentiment analysis models alone.

**Power.** Power is not a statistically significant feature in predicting empowerment. This corroborates the idea that empowerment is not the same as power – empowerment is a more nuanced and subtle concept that extends beyond power-related lexicons, relying more on the implications between

---

[6]We combined I/you and we/they because they consistently followed the same patterns. This was also motivated by qualitative analysis: I/you occurred frequently in language that was addressed directly to the other conversant, which was common in empowering posts, whereas disempowering posts often dismissed a group vaguely without addressing the conversant as an individual, leading to greater use of we/they.

the lines like the tone of the message.

**Singular vs. Plural Pronouns.** Interestingly, empowerment and disempowerment tend to use different types of pronouns. Singular pronouns (*I, you*) are positively associated with empowering language, while plural pronouns (*we, they*) are linked to disempowering language. Our manual inspections suggest one possible explanation: people who write empowering posts tend to speak directly to the listener, and also include elements of their own personal experience, hence the prevalence of *you* and *I* pronouns. Disempowering conversations are less personal and individualized, often making generalized assumptions or judgments about people.

## 4.2 Empowering Language by Gender

As a preliminary analysis of empowerment across one social dimension, we explore the differences in empowering posts written by men and women. First, we standardize the LIWC feature values for men and women's empowering language over the entire dataset. We find that women's empowering language displays significantly higher levels of positive tone and positive emotions than men. Women also use more exclamation points, while men use more swear words. These findings align with prior works in sociolinguistics that have associated exclamation points with higher expressiveness and excitability (Bamman et al., 2014; Waseleski, 2017; Güvendir, 2015), which is usually more socially acceptable for women. Meanwhile, men's use of strong or offensive language is linked with masculinity or aggressiveness, and is less socially accepted in women. Additionally, there are other features where women and men's empowering posts diverge – women use more present tense than men, and men are much less likely to use gendered words.

We then control for gender, comparing men's empowering language with all men's posts, and likewise for women. The results show that positive tone, positive emotions, and exclamation marks remain strongly correlated with empowering language even after accounting for gender. However, considering gender does impact the degree of positivity and the use of exclamation marks. Men's empowering language, when compared to men's average language, displays a greater increase in positive tone, positive emotions, and the use of exclamation marks compared to women's empowering language in relation to their average language. This suggests

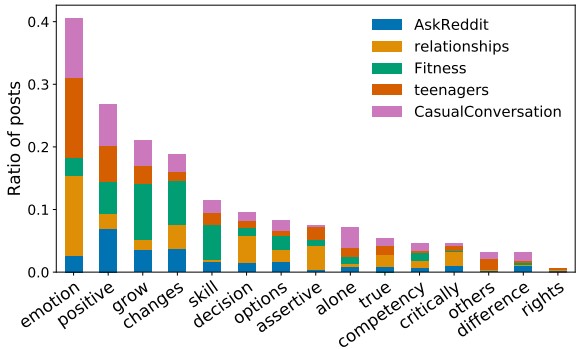

Figure 3: Distribution of empowering reasons. One post can have more than one empowering reason.

| Input Type | RoBERTa | | GPT-3 | |
|---|---|---|---|---|
| | F1 | Acc | F1 | Acc |
| Post | 63.5 | 77.7 | 36.9 | 59.7 |
| +response | 66.1 | 78.3 | 31.5 | 52.1 |
| +context | 65.5 | 77.9 | 37.5 | 64.2 |
| +all | **67.1** | **78.4** | **38.2** | **67.1** |

Table 2: Model performance of RoBERTa-based classifier fine-tuned on TalkUp and GPT-3 without fine-tuning. RoBERTA-+all is significantly better than RoBERTA-Post in terms of F1 ($p<0.1$).

that men tend to exhibit a more pronounced shift towards positive and expressive language when expressing empowerment, whereas women's empowering language already aligns closely with their overall language patterns.

Our findings highlight the complex interplay between language, gender, and empowerment, motivating future research into the influence of social factors on communication of empowerment. More detailed analyses on empowerment differences by gender and subreddit can be found in Appendix A.

### 4.3 Reasons Why Posts Are Empowering

Figure 3 illustrates the distribution of reasons selected by at least two annotators for why a post was empowering/disempowering, broken down by subreddit. The most common reasons a post was considered empowering are encouraging expression of emotions (40.6%), supporting the reader's self-image (26.8%), and supporting the reader's ability to grow (21.1%) and change (18.8%).

Notably, there are significant differences in the reasons most commonly used in different subreddits. For example, the *teenagers* and *relationships* subreddits tend to empower users by promoting expression of emotions, while empowerment in *Fitness* was more focused on encouraging people to improve themselves and make changes. The unique distributions of reasons among different communities and topics of discussion suggests that empowerment serves diverse purposes and implies different meanings depending on the context. Future work could explore which techniques should be used to empower people in specific contexts, such as empowering clients in clinical psychology or students in educational settings, based on the desired interaction goals.

### 4.4 Empowerment and Poster-Commenter Alignment

While a commenter can take either an agree, neutral, or disagree stance with the poster, most empowering posts were in conversations where the poster and commenter *agreed* (79.6%). Likewise, most disempowering posts occurred when the poster and commenter *disagreed* (45.5%). Intuitively, this makes sense for the majority of cases – people often respond agreeably to empowerment and negatively to disempowerment.

Importantly, however, this is not always the case: empowering posts can sometimes have commenters who *disagree*, and disempowering posts can have commenters who *agree*. These cases often involve more complex pragmatics. Empowering posts that contain toxic positivity are frequently met with disagreement, and sometimes commenters will reject or minimize empowering compliments for the sake of politeness. Empowerment can also be met with antagonism from an ill-intentioned commenter, regardless of how genuine the original post may be. Disempowering posts that disparage a particular group might receive an agreeing comment from someone who also shares that view of the group. We elaborate on these conversational patterns in Appendix A.3. Overall, the empowering-disagree and disempowering-agree cases provide a rich corpus for studying implicature and interactions in social contexts.

### 4.5 Modeling Empowering Language

To explore how well empowerment can be captured by computational methods, we present empowerment detection experiments with two large language models: fine-tuned RoBERTa and zero-shot GPT-3.[7] We note that our goal here is not to

---

[7]We opt to experiment with zero-shot rather than few-shot settings because fine-tuning a model of GPT-3's scale

build a state-of-the-art model, but to give a general picture of how well existing models work and to illustrate the usefulness of our dataset.

**Fine-tuned RoBERTa.** We assess how well empowerment can be identified by a pre-trained RoBERTa model (Liu et al., 2019) fine-tuned on TalkUp, and we conduct an ablation study to examine the importance of contextual information in helping the model classify a post as empowering, disempowering, or neutral. We test four model variants: post, +response (post and response), +context (post, posters' gender, subreddit), +all (post, response, context). We divide 1733 unambiguous samples from TalkUp into 60:20:20 for train:validation:test sets and select the model with best validation macro-f1 score.[8]

Table 2 presents the average macro-f1 scores across 10 separate runs using different random seeds on the test set. The results show that additional context improves model performance.

**Zero-Shot GPT-3.** Additionally, we evaluate GPT-3 Davinci's (Brown et al., 2020) ability to detect empowerment using prompts. We design seven different prompts for each of the four combinations of post+context, and generate responses. While most of GPT-3's responses are single word (e.g. "empowering"), some are longer. To map GPT-3's responses to empowerment labels, we use a simple lexical counting method: if the generated text contains more empowering-related words (e.g. empowering, empowered, empower) than words related to other labels, it is classified as empowering. GPT-3's final classification for each post takes the majority vote over its responses to the seven prompts. A full list of our GPT-3 prompts can be found in Appendix C.2.

Our results indicate that GPT-3 performs poorly in zero-shot settings compared to RoBERTa-based classifiers fine-tuned on TalkUp. This reveals that even large language models cannot effectively capture empowering language, highlighting the importance of having a carefully annotated dataset of nuanced examples like TalkUp.

---

is impractical for most users, and because our preliminary experiments indicated that few-shot prompts resulted in lower performance than zero-shot. Although in-context examples often improve performance, there are cases in which few-shot underperforms zero-shot due to models becoming excessively fixated on the provided examples and struggling to generalize effectively. This phenomenon is documented in numerous previous studies (e.g. Fei et al., 2023), and we consistently observed this in our case.

[8]Specific training details and hyper-parameters can be found in Appendix B.3.

## 4.6 Ambiguity of Empowering Language

TalkUp contains 228 samples that either were labeled as "ambiguous" by at least two annotators, or were labeled "no consensus" because all three annotators marked different answers for the empowerment question. We qualitatively analyzed this subset of TalkUp, and we find that these ambiguous posts are not "bad data," but rather are linguistically interesting *because* they are ambiguous – they are examples of language that could reasonably be interpreted in several different ways.

For example, the post "*Maybe call a relative or friend who has a car? Youll figure it out. I wish you luck, kid.*" was unanimously labelled as "empowering" and "ambiguous" by annotators. This makes sense – the post overall seems to provide a helpful suggestion, but calling the responder "kid" could be interpreted in different ways (e.g. as an endearing nickname vs. a condescending title) depending on the social relationship between the poster and the responder. Notably, many of the posts with inherent ambiguity display *sarcasm*, such as the posts "i love you too?!" and "thats grimy as f*ck but sure you do that." Sarcasm, by design, disguises a negative message in positive words, and so a sarcastic post could be interpreted either way depending on whether the sarcasm was meant positively or negatively.

We also investigated how GPT-3 handles such ambiguous cases. We find that GPT-3 tends to classify them as neutral, even for explicitly empowering posts such the above example. Instances in which the posts carried a sarcastic tone were commonly interpreted by GPT-3 as neutral as well, indicating that simultaneously empowering and ambiguous language is poorly understood by the model. The fact that ambiguity is still challenging for large models motivates the need for further work in this area, and TalkUp provides diverse examples of ambiguous language that can be used to to work towards this end.

## 5 Example Application: Unearthing Empowerment Patterns on Reddit

As a case study, we demonstrate how TalkUp and the trained empowerment classifier can be used to uncover interesting patterns in how people use empowering language. Specifically, we apply the trained classifier in §4.5 to generate empowerment labels of *all* Reddit posts and responses in RtGender, to learn about how both posters and responders

| | % Empower | | % Disempower | |
| | Post | Response | Post | Response |
|---|---|---|---|---|
| r/AskReddit | 12.0 | 14.1 | 6.8 | 5.6 |
| r/relationship | 38.7 | 27.2 | 12.7 | 11.4 |
| r/Fitness | 30.0 | 28.3 | 7.2 | 5.6 |
| r/teenager | 24.2 | 24.8 | 6.3 | 5.7 |
| CasualConversation | 25.6 | 29.2 | 2.8 | 2.3 |
| Overall | 15.2 | 16.5 | 6.9 | 5.8 |

Table 3: The percentage of empowering and disempowering posts and responses in each subreddit.

| | | Post | | Response | |
| Poster | Responder | %E | %D | %E | %D |
|---|---|---|---|---|---|
| Man | Man | 13.4 | 6.5 | 13.8 | 5.9 |
| | Woman | 16.2 | 7.1 | 18.1 | 6.0 |
| Woman | Man | 16.5 | 6.9 | 16.7 | 6.3 |
| | Woman | 20.2 | 7.3 | 20.4 | 6.4 |

Table 4: The percentage of empowering (%E) and disempowering (%D) posts and responses in RtGender classified by the model trained with TalkUp, broken down by the gender of both the poster and responder.

communicate.[9] We analyze empowering and disempowering posts in different subreddits and by different gender of the poster and responder.

**By Subreddit** Table 3 shows the percentage of empowering and disempowering posts and responses in the five subreddits of TalkUp. The results indicate that the subreddits have significantly different degrees of empowerment, and that and certain subreddits (e.g. relationship, Fitness) are significantly more empowering than others (e.g. AskReddit). Our model can be used to monitor the overall empowerment level of communities and identify unusual patterns, such as a significant rise in disempowerment. Furthermore, we find that there are more empowering responses than posts in total. On the contrary, there are more disempowering posts than responses across all subreddits. This may be because responses are often directed towards specific posts or users, and as a result, the writer may be more conscious of their tone and try to be more empowering compared to posts.

**By poster and responder gender** Table 4 shows the percentage of empowering and disempowering context by the gender of posters and responders. Overall, women seem to post and interact with

---

[9]Given that responses are only available for the posts and not for the responses, and that some samples in the data do not provide the gender of the responder, we used a model that only incorporates subreddit information as additional context to the text itself.

more empowering content. Unsurprisingly, the results show that of all the posts predicted to be empowering, women wrote a considerably higher percentage of them than men. Interestingly, however, women are also responsible for a slightly higher percentage of *disempowering* posts than men. Another surprising finding is that posts written by men that were commented on by women tend to be more empowering or more disempowering than those commented on by men, suggesting that women not only post more empowerment-charged language, but they also *engage* with more empowerment-charged posts. This may be tied to factors like the topics or types of posts that women tend to engage with and could be used to answer sociological questions about gender and social media.

## 6 Related Work

To our knowledge, Mayfield et al. (2013) is the only prior work exploring empowerment in NLP, but the contributions of our works are quite different. Mayfield et al. (2013) primarily focus on an algorithm for predicting rare classes and use empowerment as an example. In contrast, we focus on understanding empowering language itself, before developing automated detection tools. We explore the reasons behind empowerment, considering multiple dimensions of social context such as gender, topic, and poster-commenter alignment. Mayfield et al. (2013) use non-public data from a specific cancer support group, while TalkUp spans diverse topics and user bases, making our scope broader and more generalizable.

As empowering language is not well understood in NLP, our work has also drawn insights from research on related concepts:

**Power.** Danescu-Niculescu-Mizil et al. (2011) develop a framework for analyzing power differences in social interactions based on how much one conversant echoes the linguistic style of the other. Prabhakaran and Rambow (2014a,b) predict power levels of participants in written dialogue from the Enron email corpus, and several other of their works explore power dynamics in other contexts, such as gender (Prabhakaran et al., 2014b) and political debates (Prabhakaran et al., 2014a).

Our work studies *empowerment* rather than power. Power is certainly a closely related concept, but empowerment is a distinct linguistic phenomenon – it concerns not just static power levels, but interactions that *increase or decrease* a person's

power, and it is also a broader concept that encompasses things like self-fulfillment and self-esteem. While power has primarily been analyzed at the word level, such as by examining connotations of particular verbs (Sap et al., 2017; Park et al., 2021), our work attempts to look at higher-level pragmatics – implications that may not be captured by word choice alone, but suggested between the lines.

**Condescension.** The closest concept to empowerment that has been more thoroughly studied in NLP is *condescension*. Prior works have defined condescension as language that is not overtly negative, but that assumes a status difference between the speaker and listener that the listener disagrees with (Huckin, 2002). Intuitively, condescension can be interpreted as roughly the *opposite* of empowerment: it implicitly suggests that the listener has lower status or worth.

Our work particularly builds upon Wang and Potts (2019): they develop TalkDown, a dataset of Reddit posts labeled as "condescending" or "not condescending." Specifically, they identify condescending *posts* by looking for *replies* that indicate the original post is condescending. Our approach is parallel to this work: we likewise surface Reddit posts whose *responses* indicate that the *original post* is empowering (thus aligning with our definition of empowerment in §2 as an effect on the listener). TalkUp complements TalkDown by focusing on the positive aspect of such language: instead of only identifying text as condescending or not condescending, we distinguish between disempowering, empower, and neutral posts.

## 7 Future Directions

In this work, we focus only on empowerment classification and detection, with our primary contribution being the proposal of a new dataset to facilitate research in a new area of computational sociolinguistics. However, TalkUp not only can be used to detect empowerment, but also to *generate* more empowering language. As in Sharma et al. (2021b), we believe a classifier trained with our data can be used to assign rewards that tailor a generation model to produce more empowering outputs. An empowerment classifier can also be used for controllable text generation with constrained decoding, as in Yang and Klein (2021), Liu et al. (2021), and Kumar et al. (2021). Additionally, a model that can control for empowerment could be used to suggest edits to make human-written text more empower-

ing, which has potential applications in real-world dialogue settings like education and psychotherapy.

TalkUp focuses on simple two-turn interactions with 3 social variables (gender, alignment, and topic), but its framework can extend to more complex social interactions. For example, there are many other social roles that can influence power dynamics, including occupation (e.g. manager vs. employee), race (e.g. White vs Person of Color), and age (e.g. old vs. young person). Different combinations of these identities can result in further intersectional dynamics (Crenshaw, 1990; Collins and Bilge, 2020; Lalor et al., 2022). Additionally, since most real-world conversations are long back-and-forth exchanges, we encourage future work to explore empowerment in multi-turn dialogues.

## 8 Conclusion

We explore the problem of empowerment detection, grounding it in relevant social psychology and linguistics literature. To facilitate studies of empowerment, we create TalkUp, a high-quality dataset of Reddit posts labeled for empowerment and other contextual information. Our preliminary analyses demonstrate that empowerment is not captured by existing NLP methods and models, but that it can be detected with our dataset. Furthermore, we demonstrate the importance of social context in understanding empowering language with different genders, poster-commenter alignments, and topics of discussion. In studying empowerment, we work towards bigger open challenges in pragmatics, implicature, and social context in NLP.

## Ethics Statement

In constructing our study, we took precautions to ensure the task design, data collection and handling are done ethically and according to current recommended practices and guidelines (Townsend and Wallace, 2016; Mislove and Wilson, 2018; Gebru et al., 2018; Bender and Friedman, 2018). Specifically, we ensured fair compensation by calculating the pay based on minimum wage in CA (higher than then the average pay worldwide, including most U.S. states). To avoid exposing the annotators to potentially offensive or otherwise harmful content from social media, we manually checked every data sample. Beyond scientific goal of our work to understand sociolinguistic characteristics of empowering language and open new directions to NLP research on deeper pragmatic phenomena,

the practical goal is to advance NLP technologies with positive impact through understanding and incorporating empowerment in practical applications including education, therapy, medicine, and more.

## Limitations

We identify three primary limitations of our work. First, to protect the anonymity of annotators, we did not explicitly control for annotator demographics. It is thus possible that our annotator demographics is imbalanced which can impact annotation decisions and potentially incorporate biases in NLP models built on the dataset (Geva et al., 2019).

Second, with the goal to incorporate social context, we relied on gender annotations from RtGender, the corpus we draw from to annotate empowering conversations. Thus, TalkUp only centers on binary gender identities and is limited by the scarcity of data on nonbinary identities in the Rt-Gender dataset. Building resources and methods inclusive to queer identities is an important area for future work. Additionally, RtGender's gender labels were constructed by finding users who posted with a gender-indicating flair, which means that Rt-Gender only contains posts from a subset of users who voluntarily disclosed their gender; this may silence the voices of users who are less likely to share their gender, including nonbinary users. Further, future work on empowerment should incorporate broader social contexts, e.g. relationships involving inherent power hierarchies (Prabhakaran and Rambow, 2014a), more dimensions of identity like race (Field et al., 2021), and others.

Finally, TalkUp is limited to the Reddit domain and only includes English posts. This data may not be generalizable to other domains, such as clinical psychology or education.

## Acknowledgements

We gratefully acknowledge support from NSF CAREER Grant No. IIS2142739, the Alfred P. Sloan Foundation Fellowship, and NSF grants No. IIS2125201 and IIS2203097. Any opinions, findings and conclusions or recommendations expressed in this material are those of the authors and do not necessarily state or reflect those of the funding agencies.

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

## A  Empowering Language by Group

### A.1  Empowering Language by Gender

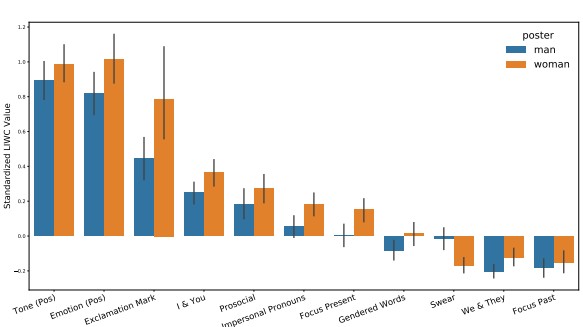

Figure 4: Average of standardized LIWC score of empowering language by men and women. The error bar indicates the 90% confidence interval.

Figure 9 illustrates the average standardized scores of empowering language by men and women.

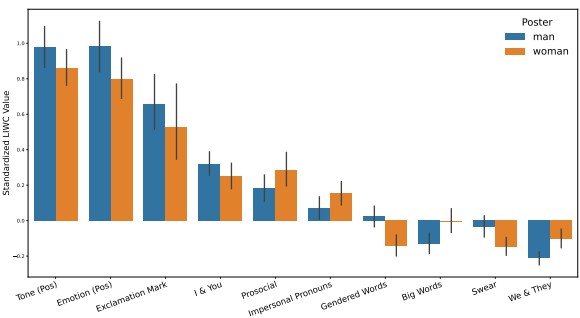

Figure 5: Average of standardized LIWC score of empowering language by men and women standardized by average of all men and women's post, respectively. The error bar indicates the 90% confidence interval.

Figure 5 illustrates the average standardized scores of empowering language by men and women after controlling for gender. In other words, we comparing men's empowering language with all men's posts, likewise for women.

### A.2  Empowering Language by Subreddit

### A.3  Empowering and Disempowering Language and Poster-Commenter Stance

**Empowering+Disagree.** Some posts labeled as empowering had commenters who *disagreed* with the poster. Figure 7 shows some notable features of these posts. Through qualitative analysis of empowering+disagree posts, we observe a few conversation patterns:

(1) Posts with *toxic positivity*, whether intentional or not, are often met with disagreement.

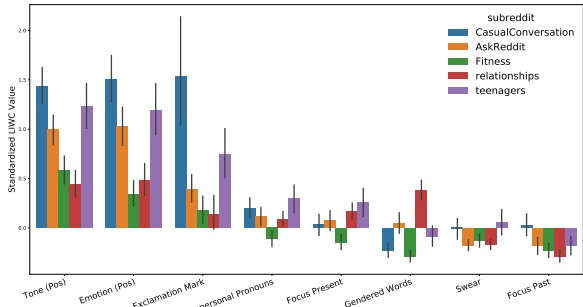

Figure 6: Average of standardized LIWC score of empowering language by subreddit. The error bar indicates the 90% confidence interval.

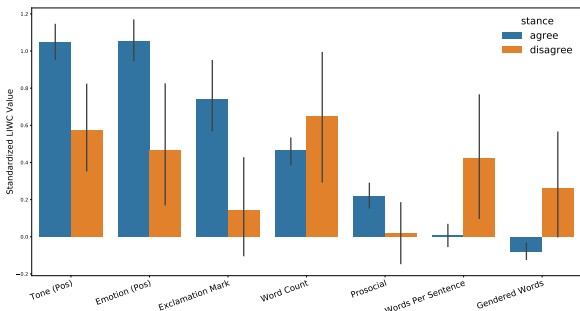

Figure 7: Average of standardized LIWC score of empowering language by stance of responder to the poster. The error bar indicates the 90% confidence interval.

Toxic positivity is a phenomenon where a positive attitude actually minimizes someone's experience, and is an open area of research (Upadhyay et al., 2022). A post with a lot of encouragement or affirmations could come across as dismissive or invalidating of the recipient's struggles.

(2) Commenters may disagree with an empowering post in an effort to be polite or humble rather than accepting the compliment. For example, one poster wrote, "That's cool!," and a commenter replied with "haha it's not as cool as it sounds." It is unlikely that the commenter actually thinks the topic of discussion is not that great; rather, rejecting compliments is a well-documented form of politeness that is most common in high-context languages (hui Eileen Chen, 2003; Gao et al., 2017). Reading between the lines to pick up on implications like this is an open area of research that involves cultural norms and values.

(3) Some empowering posts are met with antagonism from the commenter – actively attacking the poster with insults like "dummy" or "f*ck off" without really engaging in conversation. This suggests that whether or not text is perceived as empowering depends partially on the attitude and intentions of

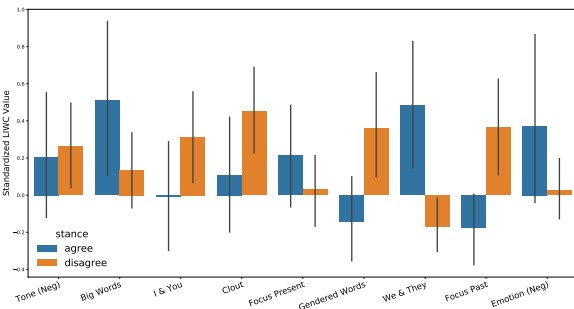

Figure 8: Average of standardized LIWC score of disempowering language by stance of responder to the poster. The error bar indicates the 90% confidence interval.

the recipient. No matter how genuine an empowering post may be, a reader may still reject it for other contextual reasons, such as being unwilling to receive feedback or simply disliking the poster.

**Disempowering+Agree.** Additionally, some disempowering posts had commenters who *agreed* with the poster. Figure 8 shows notable features of these posts, and we again inspect them qualitatively to identify two main patterns:

(1) Some posts labeled as disempowering would certainly be disparaging to a particular audience (e.g. a post that makes fun of the eating habits of vegan people would likely be received negatively by a vegan person), but the particular commenter who responded happened to share their view and joined the poster in making fun of the other group together. This is manifested in the prevalence of the *We+They* feature – such posts include many "we" and "they" pronouns because they involve the poster and commenter taking the same side and making fun of some other group.

(2) Other posts labeled as disempowering were instances where the poster was sharing very heavy or personal stories, and the commenter was validating their experience. This is exhibited particularly in the *emotion* and *tone* features: the emotion expressed in these posts is very negative because the topics themselves are heavy, but the tone is not negative because it is not negativity directed at the other person in the conversation. We note that some of these personal stories could be interpreted as neutral posts under our label definitions (i.e. the post only talks about the poster and is not relevant to the commenter), but these posts do not quite fall under this category because they were still direct conversations with the commenter. A commenter – or an annotator labeling the conversation after the fact – may feel disempowered by the contents of

such posts because empowerment has less to do with the literal words spoken and more to do with the way text impacts the feelings of the recipient, resulting in a label of "disempowering" even if the commenter is supportive of the poster.

### A.4 Ambiguous and Unambiguous Language

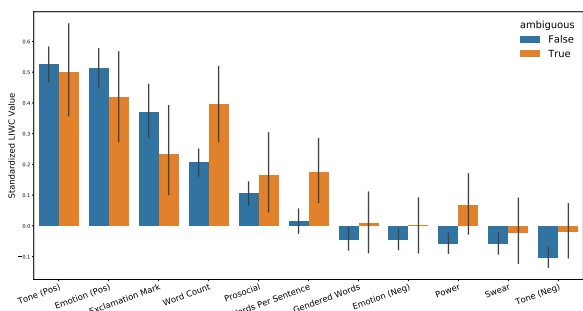

Figure 9: Average of standardized LIWC score of samples that are ambiguous and unambiguous in their empowerment. The error bar indicates the 90% confidence interval.

## B Implementation Details

### B.1 Empowerment Regression Model for Sample Selection

We trained a RoBERTa-based regression model, using the ROBERTA-BASE model on Huggingface transformers library (Wolf et al., 2020), to rank the Reddit posts to surface more likely empowering examples in the data for annotation. We used the data we collected from pilot studies to train the first model and continually updated the model as we collect more data from AMT, resulting in a total of 9 updates. The data was split into train and test set at an 8:2 ratio. In order to have float values to predict for the model, we mapped disempowering, neutral, empowering labels to 0, 0.5, 1, respectively. We only used the text of the post as an input to the model and we set maximum input length to 512. The batch size was fixed to 8. In every update, the hyper-parameters were tuned through a grid search (gradient accumulation count: {1,2,4}, warm-up ratio: {0.05, 0.1, 0.2}, learning rate: {1e-5, 1e-4, 5e-4}).

### B.2 Linear Regression Experiments

We used the statsmodels package (Seabold and Perktold, 2010) to fit a ordinary least squares linear model with intercept. Same as the RoBERTa-based empowerment regression model, we mapped empowerment labels to float values, and only used

1733 samples in TalkUp marked as unambiguous by annotators. The $R^2$ of the fitted model with features in Figure 2 was 0.29.

### B.3 Empowerment Classifier Fine-tuning

We used a ROBERTA-BASE checkpoint on Huggingface, which has 123 million parameters, to fine-tune to train a empowerment classifier discussed in §4.5. The data was split into train, development, and test sets in a 60:20:20 ratio and stratified by subreddit. We set the batch size to 32 and maximum input length to 300 even though the longest input in our data was shorter. All other hyper-parameters was set to the default values provided by the Trainer and TrainingArguments class in the transformers library. We trained each model for 10 epochs and selected the model with the best F1 score on the development set as the final model for evaluation. The model was trained on one A6000 GPU and took about 15 minutes. We ran training with 10 different random seeds and averaged the test set performance for each model.

## C Model Evaluation Details

### C.1 RoBERTa Input Type Examples

From preliminary experiments, we noticed that depending on how you format the additional input (e.g. response, subreddit, poster's gender) to RoBERTa , the performance varies. We used the input type with the best performance for each model in Section 4.5 and provide results of all templates we tried in Table 5.

### C.2 GPT-3 Prompts

As with all prompt-based language models, there is no straightforward way to determine the optimal prompt for a task, and the performance of GPT-3 can vary depending on the design of the prompt. To increase the robustness of the evaluation, we created seven templates for each model type and used the majority voting as the final output from GPT-3. We provide all templates and their corresponding performance in Table 6. While the performance of GPT-3 is not as high as the fine-tuned classifiers, practitioners can refer to this performance by template as a reference when using GPT-3 to probe empowerment in language.

## D Pilot Studies

Before crowdsourcing any data, we performed six internal pilot studies to iteratively refine our an-

notation task.[10] After each pilot, we computed annotator agreement and manually walked through every example that annotators disagreed on in order to clarify confusing aspects of our definitions. We summarize the key findings of these initial pilot studies.

**Context is useful for judging a post.** Annotator confidence was higher when we provided not just the text of the post, but additional contextual information like the poster's gender and the subreddit. Additionally, including the responder's comment helped to provide useful context by revealing how a real reader reacted to the post. Our final annotation task incorporated this contextual information.

**Annotators' free-response descriptions of social roles lack consistency.** Early iterations of our pilot studies asked annotators to specify what social group would be empowered or disempowered by a post. Answers varied dramatically – from general groups like "Democrats" to extremely specific descriptions like "a person who likes soccer and supports this sports team" – and were difficult to organize in any consistent way. However, our manual inspections of data samples revealed that most fell into two categories: (1) conversations where the poster and commenter agree/share the same stance (such as being members of the same political party or supporting the same sports team), and (2) conversations where they disagree/have opposing stances. This generalization of social relationships, while quite broad, allowed us to capture the diversity of possible social roles, and we used this stance agreement/disagreement question in the final annotation task.

**Models can help to surface potentially empowering posts.** By training a model on the pilot data collected so far, we were able to significantly increase the yield of posts that were actually labeled as empowering by annotators. To ensure we annotate a diverse range of posts, our final annotation task was done with half model-surfaced posts and half randomly-sampled posts.

**Posts are often inherently ambiguous.** Even with additional context, many posts could be reasonably interpreted as either empowering or disempowering due to inherently ambiguous linguistic phenomena like sarcasm.

| | | Input template | Test F1 |
|---|---|---|---|
| **Post** | 1 | "{post_text}" | 63.5 |
| **+Response** | 1 | "{post_text}, reply: {response_text}" | 66.1 |
| | 2 | "{post_text}{SEP_TOKEN}{response_text}" | 65.7 |
| **+Context** | 1 | "{post_text}{SEP_TOKEN}{subreddit} {poster_gender}" | 65.5 |
| | 2 | "A {poster_gender} in the {subreddit} subreddit: {post_text}" | 65.0 |
| **+All** | 1 | "A {poster_gender} in the {subreddit} subreddit: {post_text}{SEP_TOKEN}{response_text}" | 67.1 |
| | 2 | "{post_text}{SEP_TOKEN}{response_text}{SEP_TOKEN}{poster_gender} {subreddit}" | 66.6 |

Table 5: Templates used to convert additional context as a text input to the classifier. The best-performing template for each model type was used in §4.5

# E  Qualities of Empowerment

# F  AMT Details

## F.1  AMT User Interface

## F.2  Task Instruction

---

[10]These pilot studies were conducted with the authors and a small pool of computer scientists and NLP researchers.

| Input Type | F1 | Prompt |
|---|---|---|
| post-1 | 26.7 | {text}. Is this post empowering, disempowering, or neutral? |
| post-2 | 39.3 | {text}. Rate how much this text empowers people. Choose between empower, neutral, disempower. |
| post-3 | 38.8 | {text}. Which of these choices best describes the above post? Empowering, disempowering, neutral. |
| post-4 | 32.6 | Is the following post empowering, neutral, or disempowering? {text}. |
| post-5 | 25.4 | Rate whether this post is empowering, disempowering, or neutral: {text}. |
| post-6 | **39.8** | {text}. How would a reader feel about this post? Choose between empowered, disempowered, and neutral. |
| post-7 | 39.1 | Would a reader feel empowered, disempowered, or neutral after reading this post? {text} |
| +response-1 | **35.1** | Post: {text}. Response: {response}. Rate how much the post empowers the responder. Choose between empower, neutral, disempower. |
| +response-2 | 28.7 | Post: {text}. Reply: {response}. Is the post empowering, disempowering, or neutral? |
| +response-3 | 29.3 | This is a Reddit post: {text}. Someone commented: {response}. Rate whether the post was empowering, neutral, or disempowering to the commenter. |
| +response-4 | 28.5 | Is the following post empowering, neutral, or disempowering? {text}. Another user replied: {response}. |
| +response-5 | 26.4 | Rate whether this post is empowering, disempowering, or neutral. Post: {text}. Response: {response}. |
| +response-6 | 34.3 | {text}. A reader commented: {response}. How does this reader feel about the post? Choose between empowered, disempowered, and neutral. |
| +response-7 | 26.7 | Choose whether a reader would feel empowered, disempowered, or neutral after reading the following post. {text}. The reader replied: {response} |
| +context-1 | 28.9 | {text}. This post was written by a {opgender} in the {subreddit} subreddit. Is this post empowering, disempowering, or neutral? |
| +context-2 | **38.5** | {text}. This post was written by a {opgender} in the {subreddit} subreddit. Rate how much this text empowers people. Choose between empower, neutral, disempower. |
| +context-3 | 37.0 | {text}. This post was written by a {opgender} in the {subreddit} subreddit. Which of these choices best describes the above post? Empowering, disempowering, neutral. |
| +context-4 | 31.8 | If a {opgender} wrote this post in the {subreddit} subreddit, would it be empowering, disempowering, or neutral to a reader? {text}. |
| +context-5 | 37.0 | This post was written by a {opgender} in the {subreddit} subreddit. {text} Which of these choices best describes the above post? Empowering, disempowering, neutral. |
| +context-6 | 22.6 | Rate whether this post written by a {opgender} in the {subreddit} subreddit is empowering, disempowering, or neutral: {text}. |
| +context-7 | 40.3 | Would a reader feel empowered, disempowered, or neutral after reading this {opgender}'s post in the {subreddit} subreddit? {text} |
| +all-1 | 32.9 | {text}. This post was written by a {opgender} in the {subreddit} subreddit, and there was a reply: {response}. Is the post empowering, disempowering, or neutral? |
| +all-2 | 34.0 | Post: {text}. Response: {response}. The post was written by a {opgender} in the {subreddit} subreddit. Rate how much the post empowers the responder. Choose between empower, neutral, disempower. |
| +all-3 | 36.4 | {text}. This post was written by a {opgender} in the {subreddit} subreddit. Which of these choices best describes the above post? Empowering, disempowering, neutral. |
| +all-4 | 34.4 | This post was written by a {opgender} in the {subreddit} subreddit. {text} A reader responded: {response}. Which of these choices best describes the above post? Empowering, disempowering, neutral. |
| +all-5 | 34.0 | A {opgender} posted this in the {subreddit} subreddit: {text}. A reader commented: {response}. How does this reader feel about the post? Choose between empowered, disempowered, and neutral. |
| +all-6 | 20.9 | Rate whether this post written by a {opgender} in the {subreddit} subreddit is empowering, disempowering, or neutral: {text}. Someone commented: {response}. |
| +all-7 | **37.9** | Would a reader feel empowered, disempowered, or neutral after reading this {opgender}'s post in the {subreddit} subreddit? {text}. The reader replied: {response} |

Table 6: All prompts used to generate responses of GPT-3 and their macro F-1 performance over TalkUp.

| Chamberlin (1997) Element | Adapted Definition Provided to Annotators |
|---|---|
| Having decision-making power | The reader has the power to make their own decisions or influence decisions that affect them. |
| Having a range of options from which to make choices | The reader has a range of options from which to make choices. They are not restricted to only having a few limited options. |
| Assertiveness | The reader can be assertive and confidently express what they want, need, prefer, like, or dislike. |
| A feeling that the individual can make a difference | The reader feels that they can make a difference as an individual. |
| Learning to think critically; unlearning the conditioning; seeing things differently | The reader can think critically and see things from different perspectives. |
| Learning about and expressing anger | The reader can express their own emotions, like anger and sadness, in a healthy way. |
| Not feeling alone; feeling part of a group | The reader feels that they are part of a group. |
| Understanding that people have rights | The reader has rights. It can also mean broader rights, such as the right for everyone to be treated with dignity and respect. |
| Effecting change in one's life and one's community | The reader is capable of creating change in their life or community. |
| Learning skills that the individual defines as important | The reader is able to learn new knowledge and skills. |
| Changing others' perceptions of one's competency and capacity to act | The reader can change how others perceive them / their competency and capacity. |
| Coming out of the closet | The reader can come out of the closet / express who they really are. |
| Growth and change that is never ending and self-initiated | The reader can grow and change continuously and on their own volition. |
| Increasing one's positive self-image and overcoming stigma | The reader can increase their positive self-image and feel good about themselves. |

Table 7: Components of empowerment (14/15) from Chamberlin (1997) with definitions provided to annotators. The element "Having access to information and resources" was replaced with *Other* because annotators were confused by what information is excluded/included in this element during the pilot studies.

If you have not read the instructions, please read them carefully before starting the annotation task. This is a very subjective task, so some variation in answers is expected, but HITs can be rejected if the annotations clearly do not follow the instructions.

The following post was written by a **${poster}** in the **${subreddit}** subreddit:

**Post:** ${post}

**An Example of a Reader's Response:** ${response}

**Q1. Rate how empowering or disempowering the original post is to a reader.**
*Please remember that you'll be annotating the original post, not the response.*

○ Disempowering  ○ Maybe Disempowering  ○ Neutral  ○ Maybe Empowering  ○ Empowering

**Q1-2. Can this post be interpreted as either empowering or disempowering depending on what the poster meant?**
*Select Yes only if you think the post is truly ambiguous.* *If you're leaning towards empowering/disempowering, leave the answer as No.*

◉ No  ○ Yes

**Q2. If you said the post is empowering/disempowering, please select applicable reasons.**

*Definitions of these reasons are in the instructions in the sidebar.* *Leave this question blank if the post is neutral.*

☐ decision-making power          ☐ range of options to choose          ☐ can be assertive
☐ has rights                     ☐ capable of creating changes         ☐ can make a difference
☐ think critically & different   ☐ express their emotions              ☐ not alone
   perspective
☐ learn skills                   ☐ change perceptions of their         ☐ express true-self
                                    competency
☐ grow continuously              ☐ positive self-image                 ☐ others

**Q3. Does the responder agree with the poster?**

○ Disagree  ○ Neutral  ○ Agree

Submit

Figure 10: The annotation interface presented on Amazon Mechanical Turk from a worker's view.

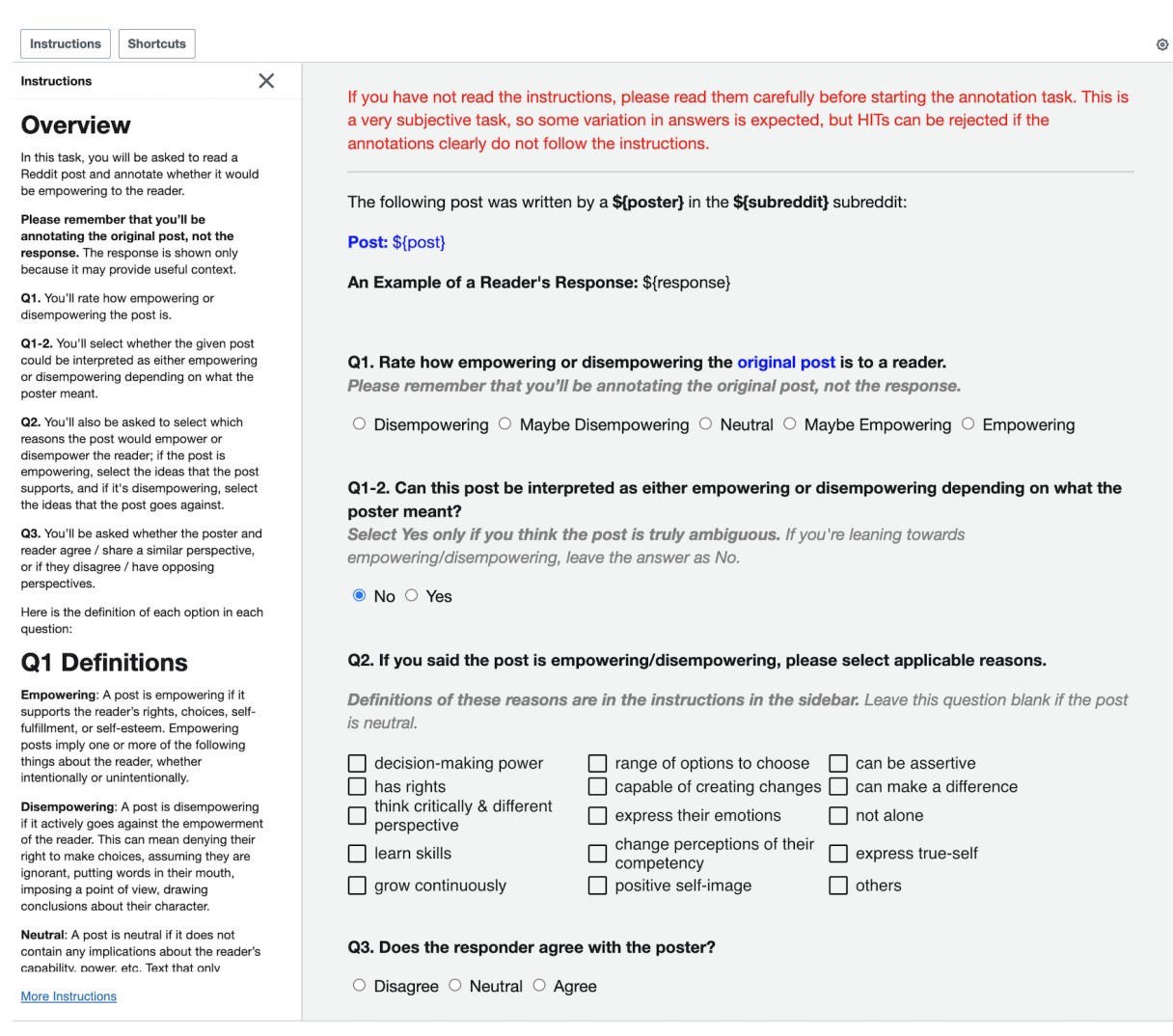

Figure 11: The annotation interface presented on Amazon Mechanical Turk from a worker's view with instruction sidebar opened.