# OpenReview forum: "TalkUp: Paving the Way for Understanding Empowering Language"
_EMNLP/2023/Conference — EMNLP 2023 Findings_

### Official Review · Reviewer_PeP9 · 2023-08-04

**Typos Grammar Style And Presentation Improvements:** Table 1 should have the number of neu…
**Soundness:** 3

**Excitement:**

3: Ambivalent: It has merits (e.g., it reports state-of-the-art results, the idea is nice), but there are key weaknesses (e.g., it describes incremental work), and it can significantly benefit from another round of revision. However, I won't object to accepting it if my co-reviewers champion it.

**Missing References:**

Works on hope speech detections are not cited or discussed : https://paperswithcode.com/task/hope-speech-detection.

**Paper Topic And Main Contributions:**

The paper introduces a new annotated dataset "talkUp" on empowerment detection in reddit posts. The data set has 2000 samples categorized into 'disempower', 'neutral' and 'empower'. It also includes the reasons for empowerment.
Further, the paper analyzes the data to find the characterestics of empowering language, relation between gender and the use of empowering language etc.
Finally, the paper offers 2 baselines for empowerment detection - a fine-tuned RoBERTa and zero-shot GPT3.

**Questions For The Authors:**

A) What is the label distributon in roberta surfaced posts vs randomly samples posts?
B) Why, how is zero-shot gpt better than provinding examples to gpt?
C) In table 2, for GPT3, why does post + response perfrom worse than simply using post?
D) Will the code be made available?

**Reasons To Accept:**

1) They release a new dataset and there are very few language resources in this domain
2) deep analysis of the intricacies of the data, including charecterestics of empowering language and its relation to gender.
3) deep analysis of the ambiguity of empowering language
4) potential uses to education and healthcare domain.

**Reasons To Reject:**

1) The inter annotator agreement is low and the reasoning for the same is not provided/convincing enough.
2) The paper claims that zero-shot GPT3 perfroms better than giving few examples in the promt (few-shot In-context learning). This is counter-intuitive and needs backing/analysis.
3) although gpt3 is impractical to fine-tune, is might be possible to fine-tune LlaMA using LoRA.
4) The data size (2000) is maybe not large enough? it could be increased and other sub-redits can be sampled.
5) The data is un-naturally skewed towards the empower category. Adding more 'disempowering' posts can balance the data. Such posts can also be surfaced using the RoBERTa.

6) possible improvement - would it be possible to have visualizaiton of roberta weights? something similar to https://github.com/jessevig/bertviz. This could strengthen the paper but not doing so in itself is not a reason to reject.

7) The question regarding ambiguity of of the post (Q1-2). The options seems inverted to me. If the post is ambiguous, the answer should be "No". However, the annotator is is asked to make "yes".

**Reproducibility:**

4: Could mostly reproduce the results, but there may be some variation because of sample variance or minor variations in their interpretation of the protocol or method.

**Reviewer Confidence:**

4: Quite sure. I tried to check the important points carefully. It's unlikely, though conceivable, that I missed something that should affect my ratings.

---

> ### Author Rebuttal · Authors · 2023-08-29
>
> Thank you for your comments. Please find our responses to your other questions below:
>
> **Annotator agreement**: Thank you for raising the concern about annotator agreement. We believe the agreement scores are actually reasonable given the complexity and nuance of this task. For empowerment annotation, we would neither expect nor want to have perfect annotator agreement – unlike many machine learning tasks, there is no “ground truth” on whether a text is empowering or not, and it’s an inherently ambiguous problem even for humans. If you compare our agreement scores with other computational social science papers on tasks of similar nature, they have similar agreement rates, especially when concerning pragmatics. For example, the dataset on latent hatred (ElSherief et al., EMNLP 2021) has a lower percentage agreement than ours, and our Fleiss’s kappa is similar to that of the Microaggression dataset (Breitfeller et al., EMNLP 2019).  We will clarify this more in detail in the paper.
>
> **Dataset size and distribution**: You pointed out that you think our dataset may be too small and skewed towards empowering language. The skew towards empowering examples was indeed by design: we want to reiterate that our data set complements the existing data set, TalkDown’s ~4000 examples (Wang and Potts, EMNLP 2019), which were skewed towards disempowering examples (explained in lines 624-638 of our paper). Due to our limited resources, the size of our dataset could not get much larger because collecting annotations for 2k examples alone cost $2000+. Thus, to build the most useful possible dataset for understanding empowering language, we focused on empowerment because TalkDown already captures a broad range of disempowerment. Such a dataset on empowering language previously did not exist, and TalkUp can be used in conjunction with a dataset of negative language like TalkDown to train models in the future.
>
> **Phrasing of ambiguity questionnaire**: We believe you might have misunderstood the phrasing in one of annotation questions (Q1-2). The question is asking if the text can be interpreted either way, which indicates its ambiguity, so annotators put “yes” to mark ambiguous cases that could not be clearly placed into empowering or disempowering without more context. None of our annotators reported an issue about the phrasing of the question; additionally, we manually reviewed all data labeled as ambiguous and verified that they made sense, so we don’t believe the quality of data was affected by how the question was phrased.
>
> **Adding LlaMA 2 after fine tuning as a baseline + Visualization of RoBERTa weights**: Thanks for these suggestions, and these may be valuable technical points that can guide future extensions of our work, but we believe that these are not within the scope of our paper to explore. As a computational social science paper, its main contribution is a new dataset that fills a gap and allows exploration of a new area of sociolinguistics – understanding characteristics of empowering language. Notably, our goal was *not* to build a state-of-the-art model: we presented results of RoBERTa and GPT3 to give a general picture of how well existing models work and to illustrate the usefulness of our proposed dataset (lines 437-441). We will clarify this more thoroughly in the paper.
>
> **GPT3 few-shot experiments**: As mentioned in the paper (footnote #8), our preliminary experiments show that few-shot underperforms the zero-shot setting, and we observe that the models become excessively fixated on the provided examples and struggle to generalize effectively to the input example. This phenomenon is well-documented in numerous previous studies (e.g., Fei et al., ACL 2023). Here’s an illustrative case: in the zero-shot setting, the correct label "empower" was obtained, whereas in the few-shot setting, an incorrect label "neutral" was assigned. This discrepancy could potentially be attributed to the sentence structure similarity between the example labeled as "neutral" and the input example. You can find the example case here: https://chat.openai.com/share/c096281b-9a85-4ab2-a2eb-84f869c922e6. Importantly, we want to highlight that this isn't a cherry-picked example; we consistently observed this pattern, and quantitatively, the few-shot approach consistently underperformed compared to the zero-shot approach. We can add more clarification about this in footnote #8; however, we would like to reiterate that this is a computational social science paper, and the details of GPT’s few-shot vs zero-shot performance are not important to the main contributions of this paper.
>
> **Will code be made publicly available?** Yes, both code for experiments and the trained classifier weights will be made publicly available.

---

### Official Review · Reviewer_8biv · 2023-08-06

**Soundness:** 4

**Excitement:**

4: Strong: This paper deepens the understanding of some phenomenon or lowers the barriers to an existing research direction.

**Paper Topic And Main Contributions:**

This paper dissects empowering language from the perspective of sociolinguistics. The authors crowdsourced a dataset from Reddit labeled for empowering. They analyzed the characteristics of empowering and disempowering language at the data level, and demonstrated the insufficient capabilities of existing models (e.g., GPT-3) at the model level.

**Reasons To Accept:**

(1) Propose a novel research point, i.e., empowering language.

(2) Provide a dataset annotated with empowerment, i.e., TalkUp.

(3) Conduct comprehensive analyzes and make a step for future research.


**Reasons To Reject:**

(1) This paper only analyzes empowerment based on gender. Other attributes, e.g., age, culture, etc., also affect the expression of empowerment.

(2) This paper only focuses on the empowerment in a single-turn post-response setting, and the empowerment in multi-turn dialogue may be more complicated when it involves more context.


**Reproducibility:**

3: Could reproduce the results with some difficulty. The settings of parameters are underspecified or subjectively determined; the training/evaluation data are not widely available.

**Reviewer Confidence:**

3: Pretty sure, but there's a chance I missed something. Although I have a good feel for this area in general, I did not carefully check the paper's details, e.g., the math, experimental design, or novelty.

---

> ### Author Rebuttal · Authors · 2023-08-29
>
> Thank you for your insightful  comments and valuable  suggestions. Please find our responses to your feedback below:
>
> **More social attributes**: We agree that it’s important to consider many aspects of social context, and this was indeed a major motivation for our project – very few works (and datasets) in NLP contain *any* social information at all, so we push in this direction by releasing a dataset that contains not just language, but 3 social factors as well (gender, alignment, and topic). A central point of our paper is that empowering language cannot be understood in isolation and can only be captured in the context of several other social variables. We didn’t have access to other demographic information due to the source of our data, and it’s impossible for any paper to take every possible demographic factor into account. But exploring other sociocultural variables is a rich area for future work, and we strongly advocate for more exploration into intersectionality / ways that multiple demographic factors can influence the pragmatics of an interaction. We intend TalkUp to be a first step in this direction.
>
> **Extension to multi-turn dialogues**:  Thanks for your great suggestion – we agree that, in the big picture, studying multi-turn dialogues is important for understanding empowering language. In this paper, as an initial exploration into an area with little to no prior work, we simplified conversations in an effort to construct the most useful framework / dataset possible with our limited access to resources. Multi-turn dialogue would have introduced many practical challenges: we would need annotators to converse with each other for several turns (which is much more complicated and costly than having them evaluate existing conversations) and would need to collect annotator demographic information (which raises privacy concerns). Focusing on two-turn dialogue allowed us to reuse existing datasets such as TalkDown and RtGender, and establish a (simplified) baseline understanding of empowering language, and we hope it lays the groundwork for future explorations into longer conversations. We will incorporate this discussion into the paper.

---

### Official Review · Reviewer_83gy · 2023-08-07

**Soundness:** 4

**Excitement:**

3: Ambivalent: It has merits (e.g., it reports state-of-the-art results, the idea is nice), but there are key weaknesses (e.g., it describes incremental work), and it can significantly benefit from another round of revision. However, I won't object to accepting it if my co-reviewers champion it.

**Paper Topic And Main Contributions:**

The paper presents a new dataset, named TalkUp, designed for identifying empowerment in internet community posts. Reddit posts are annotated to determine if they are empowering or not, the type of empowerment expressed, and the agreement of responses to the posts. The authors analyze linguistic features, gender-related aspects, and reasons to empowering language in the dataset. Baseline models are employed to evaluate the performance of identifying empowerment in the posts. Additionally, the authors propose an application to explore patterns of empowerment in the Reddit community.

**Questions For The Authors:**

-	Hope to listen to answer about the concern.

**Reasons To Accept:**

-	This paper introduces a compelling dataset and novel tasks related to empowering language.
-	The analysis of empowering language patterns from the dataset and the proposed application approach provide valuable insights for future research using the new dataset.


**Reasons To Reject:**

Considering the current trend of generation tasks in this area, it would be interesting to explore the usefulness of the new dataset for tasks such as generating empowering posts and responses. While the paper does not delve into these applications, it would be beneficial to discuss and present potential results for these tasks in future research.

**Reproducibility:**

5: Could easily reproduce the results.

**Reviewer Confidence:**

3: Pretty sure, but there's a chance I missed something. Although I have a good feel for this area in general, I did not carefully check the paper's details, e.g., the math, experimental design, or novelty.

**Typos Grammar Style And Presentation Improvements:**

It would be better to mention the Tables in the Appendix. For example, line 150 mentions ‘Appendix E’, but there is no words in the section. I guess Table 7 is the appropriate contents, so it would be better to refer the Table 7 in the section.

---

> ### Author Rebuttal · Authors · 2023-08-29
>
> Thank you for your great suggestion about mentioning the usefulness of our dataset for generation tasks; we agree and certainly envision TalkUp being useful for future applications by helping models generate more empowering language, and by generating suggestions to make human-written text more empowering. Some concrete examples we have in mind are:
>
> 1. Rewards Model for Reinforcement Learning: As in (Sharma et al., WWW  2021), we believe a classifier trained with our data can be used to assign rewards that can tailor a generation model to be more empowering
> 2. Controllable Text Generation with Constrained Decoding: As in (Yang and Klein, NAACL 2021; Liu et al., ACL 2021; Kumar et al., NeurIPS 2021), our classifier can be used to provide signals for controllable inference from pretrained generation models during the decoding process.
>
> In fact, we are already conducting a follow-up study that uses TalkUp for controllable text generation to make dialogue more empowering in real-world conversational settings, such as education and therapy. Our hope is that TalkUp will enable further progress in both understanding and generating empowering language. We will add discussion of these potential applications and future research directions in our camera-ready version.

---

### Meta-Review · Area_Chair_DykP · 2023-09-19

**Recommendation:** 4

**Metareview:**

Reviewers agreed that the core contribution of this paper - the new dataset and novel tasks presented regarding the characteristics of empowering language - was strong and compelling. The CSS analyses of this data presented in the paper were characterized by reviewers as "deep" and "comprehensive," providing "valuable insights for future research."

Reviewers presented some issues for requiring clarification and noted areas of possible future work (such as addressing other aspects of social context beyond gender, multi-turn dialogue, comparisons with TalkDown, etc); authors responded thoroughly to reviewer concerns and noted ongoing future work that addresses these considerations, ultimately resulting in reviewer consensus that the work is sound and merits some excitement.

Overall reviewers found that this work makes a meaningful contribution worthy of dissemination.

---

### Decision · Program_Chairs · 2023-10-07

**Decision:**

Accept-Findings

**Comment:**

Reviewers agreed that the core contribution of this paper - the new dataset and novel tasks presented regarding the characteristics of empowering language - was strong and compelling. The CSS analyses of this data presented in the paper were characterized by reviewers as "deep" and "comprehensive," providing "valuable insights for future research."

Reviewers presented some issues for requiring clarification and noted areas of possible future work (such as addressing other aspects of social context beyond gender, multi-turn dialogue, comparisons with TalkDown, etc); authors responded thoroughly to reviewer concerns and noted ongoing future work that addresses these considerations, ultimately resulting in reviewer consensus that the work is sound and merits some excitement.

Overall reviewers found that this work makes a meaningful contribution worthy of dissemination.